Convolutional neural network-based ensemble methods to recognize Bangla handwritten character

Shibly Mir Moynuddin Ahmed 2016-3-60-057@std.ewubd.edu
Tisha Tahmina Akter
Tani Tanzina Akter
http://orcid.org/0000-0001-7202-1087 Ripon Shamim dshr@ewubd.edu
Department of Computer Science and Engineering, East West University , Dhaka , Bangladesh
Zhou Huiyu
Electronic publication date: 2021 Jun 28
Publication date: 2021
Volume: 7
Electronic Location ID: e565
Received 2021 Jan 22; Accepted 2021 May 6
Copyright: © 2021 Shibly et al.
Copyright year: 2021
Copyright holder: Shibly et al.
License: This is an open access article distributed under the terms of the Creative Commons Attribution License, which permits unrestricted use, distribution, reproduction and adaptation in any medium and for any purpose provided that it is properly attributed. For attribution, the original author(s), title, publication source (PeerJ Computer Science) and either DOI or URL of the article must be cited.
License URL: https://creativecommons.org/licenses/by/4.0/

Keywords: Convolutional neural network, Ensemble learning, Bangla handwritten character recognition, Deep learning, Stacked generalization, Bootstrap aggregating, Image classification, Feature extraction

Funding: The authors received no funding for this work.

==============================
In this era of advancements in deep learning, an autonomous system that recognizes handwritten characters and texts can be eventually integrated with the software to provide better user experience. Like other languages, Bangla handwritten text extraction also has various applications such as post-office automation, signboard recognition, and many more. A large-scale and efficient isolated Bangla handwritten character classifier can be the first building block to create such a system. This study aims to classify the handwritten Bangla characters. The proposed methods of this study are divided into three phases. In the first phase, seven convolutional neural networks i.e., CNN-based architectures are created. After that, the best performing CNN model is identified, and it is used as a feature extractor. Classifiers are then obtained by using shallow machine learning algorithms. In the last phase, five ensemble methods have been used to achieve better performance in the classification task. To systematically assess the outcomes of this study, a comparative analysis of the performances has also been carried out. Among all the methods, the stacked generalization ensemble method has achieved better performance than the other implemented methods. It has obtained accuracy, precision, and recall of 98.68%, 98.69%, and 98.68%, respectively on the Ekush dataset. Moreover, the use of CNN architectures and ensemble methods in large-scale Bangla handwritten character recognition has also been justified by obtaining consistent results on the BanglaLekha-Isolated dataset. Such efficient systems can move the handwritten recognition to the next level so that the handwriting can easily be automated.

Introduction

Bangla is one of the prestigious languages all over the world. About 230 million people from around the world speak Bangla as their native language (Khan, Al Helal & Ahmed, 2014), and approximately 37 million people use it as a second language for both speaking and writing purposes. Thus, by the total number of speakers worldwide, Bangla is the fifth most spoken native language (Alom et al., 2018) and the seventh most spoken language as well. As Bangla is such a renowned language, works related to Bangla language such as Bangla handwritten character recognition is getting more attention among machine learning practitioners.

In this age of artificial intelligence and automation, the prominent applications of Bangla handwritten recognition cannot be overlooked. It can play a significant role in many aspects such as post-office automation, national ID number recognition, parking lot management system, and online banking (Alom et al., 2018). This recognition system can also play an essential part in signboard translation, digital character conversation, keyword spotting, scene image analysis, text-to-speech conversion (Manoharan, 2019), meaning translation, and most notably in Bangla optical character recognition (OCR) (Manisha, Sreenivasa & Sundara Krishna, 2016). But it has been a great challenge to provide such a system for Bangla than most other languages. Bangla has a very complex and rich handwriting pattern as opposed to the simple handwriting pattern of other renowned languages. In Fig. 1, a complexity comparison of a Bangla handwritten character has been illustrated i.e., Fig. 1B, a Bangla handwritten character, has been compared with an English handwritten character i.e., Fig. 1A, and also with an Arabic handwritten character i.e., Fig. 1C. From the figure, it is apparent that Bangla characters have a more complex structure than Arabic and English characters.

Figure 1 Complexity comparison of Bangla handwritten character with handwritten characters from other languages.

This figure compares the complexity of handwritten characters of Bangla with other languages. (A) An English handwritten character, (B) a Bangla handwritten character, and (C) an Arabic handwritten character.

Bangla script consists of 11 vowels and 39 consonants. These 50 characters are the basic alphabets of the Bangla language. In addition to that, there are more than 170 (Das et al., 2014) conjunct-consonant characters that are formed by combining two or more than two basic characters, and these compound characters have a very close resemblance with each other. For having very complex shaped cursive characters, morphological complexity, the variety of writing styles, and scarcity of the complete Bangla handwritten dataset, recognizing Bangla handwritten characters has become more challenging for a system. To deal with these emerging challenges, researchers have utilized many different methods such as deep convolutional neural networks (DCNNs) (Alom et al., 2018), convolutional neural network (CNN) with transfer learning (Reza, Amin & Hashem, 2020), and ensemble learning (Rahaman Mamun, Al Nazi & Salah Uddin Yusuf, 2018).

For the complex patterns that lie into the handwritten Bangla characters, their recognition is a very difficult task. There are various techniques that the researchers have followed in Bangla handwriting recognition challenges. However, convolutional neural network (CNN)-based architectures have proved to be the de-facto standard in this domain of handwritten character recognition (Rahman et al., 2015; Azad Rabby et al., 2018; Chatterjee et al., 2020), and many more. In this study, different convolutional neural networks have been utilized for Bangla handwritten character recognition. On the other hand, ensemble learning is a special method where a model is built with the help of different base classifiers to improve the quality of predictions and overall performance. The ensemble learning methods have been utilized widely for image recognition and classification for a while to achieve better performance. Many studies have explored the possibilities of ensemble methods in image analysis (Ju, Bibaut & Van der Laan, 2018; Das et al., 2018). In light of other works for obtaining outstanding results, in this study, various ensemble methods have also been used for classifying Bangla handwritten characters.

This study has developed good performing machine learning models to classify Bangla handwritten characters efficiently. The key research contributions of this study are:Developing seven convolutional neural network (CNN)-based models to classify isolated Bangla handwritten characters efficiently.

This is one of the first reported works that utilize various ensemble methods to recognize Bangla handwritten characters.

Classifying more Bangla handwritten characters than other related state-of-the-art existing works with better performances.

In the following section of this article, an extensive literature review of the related works is presented. After that, the methods, materials, and experimental framework of this study are subsequently discussed. Then, the results are explained with appropriate discussion, and finally, concluding remarks are described.

Related Works

Alom et al. (2018) have introduced the popular deep convolutional neural networks to recognize Bangla handwritten characters. They have performed their experiments on CMATERdb (Sarkar et al., 2012). The used architectures include DenseNet (Huang et al., 2017), All-Conv Net (Springenberg et al., 2015), VGG Net (Simonyan & Zisserman, 2015), FractalNet (Larsson, Maire & Shakhnarovich, 2019), ResNet (He et al., 2016), and NiN (Lin, Chen & Yan, 2014). In another work (Chatterjee et al., 2020), to resolve the issue of a higher number of iterations, the authors have used transfer learning with the ResNet50 network in a need of proper training of the model. To make the training faster, a one cycle policy and the varying image sizes methods have been applied as well. An accuracy of 94.3% at the character level has been achieved by Bhattacharya et al. (2016). In Azad Rabby et al. (2018), two datasets have been used for experiments using a CNN-based architecture. An approach of Xception-based ensemble learning has been adopted by Rahaman Mamun, Al Nazi & Salah Uddin Yusuf (2018) to recognize handwritten Bangla digits.

Rahman et al. (2015) have developed a CNN-based model to classify the basic handwritten characters. Alif, Ahmed & Hasan (2018) have modified ResNet18 architecture by adding a dropout layer for handwritten classification. Further, in Chowdhury et al. (2019) CNN-based work has been proposed by the authors. Shopon, Mohammed & Abedin (2017) have proposed a CNN-based model to classify only Bangla handwritten digits. Also, a study of handwritten digit classification has been conducted by Sharif et al. (2017). Apart from CNNs, Bangla handwritten characters have been classified using a harmony search algorithm (Sarkhel, Saha & Das, 2015). The modified quadratic discriminant function (MQDF) method has also been used for the recognition task (Pal, Wakabayashi & Kimura, 2007). A deep network has been used in this domain-specific classification as well (Sazal et al., 2013). This technique varies from the conventional methods of character preprocessing for the creation of handcrafted characteristics such as loops and strokes.

Various research have also been conducted for the recognition of handwritten characters and texts from other languages such as recognizing Baybayin scripts using SVM (Sitaula, Basnet & Aryal, 2021), analyzing handwritten Hebrew document (Biller et al., 2016), recognizing English handwritings (Pham et al., 2020), and many more. For English handwritten digit and character recognition tasks, CNN-based architectures have yielded better performance than other techniques (Baldominos, Saez & Isasi, 2019; Ranzato et al., 2007; Cireşan et al., 2011), and so on. The authors have applied different variants of CNN on the MNIST dataset and obtained good results. Similarly, good results have been obtained on the EMNIST dataset using convolutional neural networks (Peng & Yin, 2017; Mor et al., 2019).

So far, the described related works for various handwritten character recognition are mostly CNN-based. However, one of the objectives of this study is to explore the usability of ensemble methods for the commenced recognition task. Although the ensemble techniques have not been used vastly for handwritten classification like CNN, they have been used widely in other image classification tasks. Shibly, Tisha & Ripon (2021) have used the stacked generalization method for handwritten character recognition. Before this, SVM based stacked generalization has also been used for image classification (Tsai, 2005). The stacked ensemble generalization approach has already demonstrated the promising performance of disease detection in the medical field (Rajaraman et al., 2018). It has also been used for multiclass motor imagery-based brain-computer interfaces (Nicolas-Alonso et al., 2015). The other ensemble methods like bagging (Hothorn & Lausen, 2003), boosting (Fidalgo et al., 2018), random forest (Gislason, Benediktsson & Sveinsson, 2006) have also been implemented in image classification. These ensemble methods have been utilized to improve the performances of the classifiers.

The extensive literature review suggests that the convolutional neural network-based architectures are widely used in the recognition of Bangla handwritten characters. But the number of characters recognized by most of the works is limited. This study is an attempt to recognize as many handwritten characters as possible with widely used CNN architectures. Moreover, the literature review also suggests that there is a gap of knowledge about the applicability of ensemble methods in handwritten Bangla character recognition tasks. This study also explores few ensemble methods to improve the recognition performance.

Methods

In this section of the article, the methodologies of this study are presented with appropriate explanation. The three phases of the applied methodology are described in this section. A high-level overview of the workflow of this study is illustrated in Fig. 2.

Figure 2 Overview of the workflow of the study.

Convnet architectures

Six popular convnet architectures have been used to classify Bangla handwritten characters. These architectures are AlexNet (Krizhevsky, Sutskever & Hinton, 2017), VGG16 (Simonyan & Zisserman, 2015), VGG19 (Simonyan & Zisserman, 2015), ResNets (He et al., 2016), Xception (Chollet, 2017), and DenseNet (Huang et al., 2017). Additionally, a VGG-like small convnet has also been developed by us. Each architecture has its unique style, and they try to solve the image classification problem in their own different ways.

The convnets are very useful for extracting features from images. Handcrafted features for image classification are not efficient enough to build robust classifiers. With the help of convolution, which is an attempt to mimic how the brain convolves, a convnet model identifies various shapes that lie in an image. As the final step of convolution, a feature vector of fixed length for every image is generated. After that, those features are used to classify the images. The convnets can also have classification layers on top of the feature extraction layers. For having both feature extraction and classification ability integrated into a single architecture, a convnet is convenient to use. Convolutional neural network-based models have also been proved to be better performers in image classification and recognition. For these reasons, different convnets have been applied for the recognition task. Most importantly, a wide range of convnets has been used to find the optimal model for Bangla handwritten character recognition task.

AlexNet, VGG16, VGG19, small CNN

AlexNet, VGG16, VGG19, and our developed small CNN are four convolutional neural network models that have similar sequential structures. These four convolutional neural networks have been used for recognition tasks for their simple yet very powerful architectures. Among them, AlexNet is one of the very first convnet architectures that utilize graphical performance units (GPU) for processing images. The comparative description of four architectures is presented in Table 1. AlexNet consists of eight layers. The architecture starts with a convolutional layer followed by a max-pooling layer. This convolutional—max-pooling layer combination repeats two times. After that, there are three convolutional layers and a single max-pooling layer. The last three layers of AlexNet are dense while the last one being the output layer. VGG16, VGG19, and small CNN have an AlexNet-like structure. However, the VGG-networks are deeper than AlexNet. VGG16 and VGG19 are sixteen- and nineteen-layers architectures, respectively. On the other hand, the small CNN architecture has eight layers.

Table 1 Details of AlexNet, VGG16, VGG19 and a small CNN architectures.

AlexNet	VGG16	VGG19	Small CNN	
[Conv2D,3×3,96MaxPooling,2×2Conv2D,3×3,128Conv2D,3×3,192Conv2D,3×3,256Conv2D,3×3,128MaxPooling,2×2]	[Conv2D,3×3,32Conv2D,3×3,32MaxPooling,2×2Conv2D,3×3,64Conv2D,3×3,64MaxPooling,2×2Conv2D,3×3,128Conv2D,3×3,128Conv2D,3×3,128MaxPooling,2×2Conv2D,3×3,175Conv2D,3×3,175Conv2D,3×3,175MaxPooling,2×2Conv2D,3×3,225Conv2D,3×3,225Conv2D,3×3,225MaxPooling,1×1]	[Conv2D,3×3,32Conv2D,3×3,32MaxPooling,2×2Conv2D,3×3,64Conv2D,3×3,64MaxPooling,2×2Conv2D,3×3,128Conv2D,3×3,128Conv2D,3×3,128Conv2D,3×3,128MaxPooling,2×2Conv2D,3×3,192Conv2D,3×3,192Conv2D,3×3,192Conv2D,3×3,192MaxPooling,2×2Conv2D,3×3,256Conv2D,3×3,256Conv2D,3×3,256Conv2D,3×3,256MaxPooling,2×2]	[Conv2D,3×3,50Conv2D,3×3,75MaxPooling,2×2Conv2D,3×3,125MaxPooling,2×2Conv2D,3×3,175MaxPooling,2×2Conv2D,3×3,225MaxPooling,2×2]	
Flatten	Flatten	Flatten	Flatten	
1st Dense layer	1st Dense layer	1st Dense layer	1st Dense layer + Dropout (50%)	
2nd Dense layer	2nd Dense layer	2nd Dense layer	2nd Dense layer + Dropout (30%	
Softmax output layer	Softmax output layer	Softmax output layer	Softmax output layer	
Note:

The layers descriptions are given for each architecture. For convolutional layer (Conv2D), the dimension of the filter at a convolutional layer e.g., 3 × 3 and number of filters e.g., 96 are given. For max-pooling, the pool size e.g., 2 × 2 is mentioned.

ResNets

Residual Networks i.e., ResNets (He et al., 2016) have also been used for the classification challenges. The convolutional and the identity blocks are two building blocks of a ResNet that have been presented in Fig. 3. The ResNets are unique for their residual or skip connections. The information propagates from a specific layer to another specific layer following two paths. In a convolution block, the information goes through a series of convolutional—batch normalization layers. The information goes through a point-wise convolutional layer as well. Then the weights learned from the two paths are added after the end of the convolutional block. Similar things happen in identity block. To solve a sophisticated classification task, a deeper network is needed. But deeper networks tend to fail in learning due to degradation problems. Even sometimes the performance of a network falls due to adding new layers. Various skip connections in ResNets have been employed to prevent this degradation within a network.

Figure 3 Two building blocks of ResNet architecture.

The ResNet architecture is built with the help of these two blocks. (A) Convolutional block and (B) identity block.

A ResNet can be of 18, 34, 50, 101, or 152 layers. All the ResNets start with a 7 × 7 convolutional layer with 64 filters and strides of two followed by a batch normalization layer and a 3 × 3 max-pooling layer having strides of two. After that, for ResNet50 architecture, there is a convolutional block is followed by two identity blocks. Then there is another convolutional block followed by three identity blocks. This pattern continues two more times except with five identity blocks and two identity blocks, respectively. After the last identity block, the outputs are reduced with average pooling, and finally, a SoftMax classification layer is added.

Xception

There are skip connections in the Xception architectures as well. Inception network (Szegedy et al., 2015) inspired the creation of this architecture. The modification of this architecture over the Inception is that each inception module has been replaced with a depthwise separable convolutional layer. Just like a typical convolutional layer, the depthwise convolutional layer has the same output. But to reach the same output, a separable convolutional layer requires fewer computations than the regular convolutional layer. Such improvement over the convolutional layer makes a model be trained faster.

In Fig. 4, the Xception architecture is presented. Xception begins with two typical convolutional layers having 32 and 64 filters, respectively with a fixed filter size of 3×3. This is followed by five Xception blocks. Each input of such a block is passed through two separable convolutional layers followed by a max-pooling layer and also passed through a pointwise convolution via shortcut connection except the fourth Xception block. The fourth Xception block consists of only one separable convolutional layer, and this block repeats eight times. After the last Xception block, there are two separable convolutional layers followed by a global average pooling layer. After that, the fully connected layers are added with the last being the output layer.

Figure 4 Xception architecture details.

DenseNet

Another used convnet in this work is DenseNet. The idea behind this model is that every convolutional layer in a certain block is connected to every other convolutional layer of that block. Every layer not only processes the input forwarded by the immediately preceding layer but also processes the input combination forwarded by all the previous layers. Basically, a DenseNet model consists of a few dense blocks. Before every dense block, there is a convolutional layer followed by an average-pooling layer except the first dense block. The block diagram of DenseNet is displayed in Fig. 5. There is a convolutional layer with a filter size of 7×7 followed by a max-pooling layer with a pool size of 3×3 before the first dense block. The other transitional convolutional layers have a filter size of 1×1. After the last dense block, there is a global average-pooling layer and few customizable classification layers.

Figure 5 Overview of DenseNet architecture.

Image data augmentation

It has been proved that image data augmentation helps a classifier to recognize images more accurately (Perez & Wang, 2017). Here, while creating the individual CNN models, different types of data augmentation have been applied to the training images. The images have been rotated from 9 degrees to 15 degrees. The height and width shift range also has been adjusted from 0.09 to 0.15, and the zoom range is varied from 0.09 to 0.15. These types of augmentation provide more generalization to the classifiers. Different image augmentations used in this study are presented in Table 2 considering A1 as rotation range, A2 height shift range, A3 as width shift range, and A4 as zoom range.

Table 2 Different image augmentation hyperparameters.

Name	Augmentation hyperparameters	
aug0	No image augmentation	
aug1	A1=102A=0.13A=0.14A=0.1	
aug2	A1=92A=0.093A=0.094A=0.09	
aug3	A1=112A=0.113A=0.114A=0.11	
aug4	A1=10	
aug5	A1=152A=0.153A=0.154A=0.15	
aug6	A1=112A=0.113A=0.114A=0.10	
aug7	A1=132A=0.093A=0.14A=0.11	
aug8	A1=142A=0.13A=0.14A=0.1	
aug9	A1=152A=0.113A=0.114A=0.11	

Convnet as feature extractor

The described convnets can be used as feature extractors as well. The feature extractor model can be developed by removing the fully connected layers from a convnet. The flatten layer after the last pooling layer produces feature vectors. These can be used to train a separate classifier. Among the described architectures, ResNet50 has shown the highest accuracy in the classification task. This convnet has been used as a feature extractor in this work to train separate classifiers with traditional machine learning algorithms, such as Logistic Regression, Decision Tree, Naïve Bayes, and Support Vector Machine. Using a pre-trained ResNet50 feature extractor, features from train and test images have been extracted. The flatten layer of the model produces a feature vector size of 2,048. Fitting a classifier on this huge feature space is not feasible due to a lack of computational resources. To overcome this issue, three dense layers with 1,024, 512, and 80 neurons have been added before the final classification layer. This has been done to narrow down the feature space to a manageable size.

Ensemble methods

The classification task has also been carried out using some ensemble techniques. In ensemble learning, a prediction is made based on multiple learning algorithms rather than single learning algorithms (Opitz & Maclin, 1999). It is a technique used in machine learning fields where more than one model is trained for the same task as opposed to a typical machine learning technique for solving a particular task. In ensemble learning, hypotheses from different models are combined to create a more generalized, accurate, and robust model to solve a specific problem. Most commonly used ensemble techniques are majority voting, weighted majority voting, Borda count, bagging, boosting, stacked generalization, and so on. One of the objectives of this study is to explore the potential of ensemble methods in Bangla handwritten character recognition. Regarding this, in this study, stacked generalization, bootstrap aggregating, adaptive boosting, extreme gradient boosting, and random forest ensemble methods have been used for the handwritten recognition task.

Stacked generalization

Stacked generalization follows a very general process. The mechanism of this method is illustrated in Fig. 6. In this method, on the training dataset, k different first-level classifiers are fitted. There are many methods to obtain these first-level classifiers (Aggarwal, 2015). In this work, a 10-fold cross-validation method has been employed on a training set, and ten first-level classifiers have been developed. The data augmentation technique for each first-level classifier has been presented in the figure. After obtaining first-level classifiers, they are stacked, and their outputs are concatenated. After the concatenation, three dense layers have been added. The last layer of the second-level stacked model is the output layer. After creating the stacked model, it has been trained with the validation set, where the output of each first level classifier works as the new dataset for training the second level classifier. This is a lesser-used method for image classification with great potential. To explore its usability and to improve recognition performance, this method has been applied.

Figure 6 The working procedure of stacked generalization ensemble method.

Bootstrap aggregating

The second ensemble method that has been employed is bootstrap aggregating. It is also known as bagging. In bagging, from training data, a portion of images are selected randomly and put in a bag with replacement. Usually, 1−1/e=63.2% instances from the training set go to a bucket with the other 36.8% of the copied instances from randomly selected data. By doing this, a bag with the size of the original training set is constructed. This is the principal idea behind bootstrap aggregating. In Fig. 7, the overall working procedure of this method has been demonstrated along with the data augmentation technique applied on each classifier. After creating ten bags from the training dataset, an independent classifier has been fitted on each bag. And for testing, majority voting has been applied to the predictions made by individual classifiers. This method is an attempt to reduce variance from the classifier (Aggarwal, 2015). Like stacked generalization, this method has also been used to achieve better performance.

Figure 7 The working procedure of bootstrap aggregating ensemble method.

Boosting and random forest

Two boosting methods have been used in this work—adaptive boosting and extreme gradient boosting. In AdaBoost, the classification starts with equal weight to each of the training instances. The weight associated with an instance indicates the probability of being chosen in bootstrap for training on a certain iteration. If an instance is misclassified, the weight associated with that instance increases for the next iteration. If an instance is correctly classified, the weight gets decreased. The iteration of training repeatedly terminates when the classifier's accuracy becomes 100% or the classifier performs worse than the base classifier (Aggarwal, 2015). Another boosting method is XGBoost (Chen & Guestrin, 2016). It is a tree-based ensemble technique where the errors are minimized by taking extreme measures using a gradient descent algorithm.

The last ensemble technique that has been used is the random forest. A random forest is nothing but a combination of many decision trees. In this method, k decision trees are created using training data to form a forest, and for the test data, the majority voting technique is followed. The majority class predicted by individual decision trees is the predicted class. The trees are not developed with all features. Another way to create a random forest is to use a bootstrap aggregating technique (Han, Kamber & Pei, 2012). A portion of the training data points are selected with replacement, and for each decision tree, a newly sampled training dataset is used. Random forest resolves the problem of overfitting in decision tree classifiers (Vinet & Zhedanov, 2010).

Experimental Setup

This study aims at classifying Bangla handwritten characters. For creating the convnets, Keras (“Keras” Available at https://keras.io/) on top of TensorFlow (“TensorFlow” Available at https://www.tensorflow.org/)—a python library has been used. For a few of the experiments, the sci-kit-learn (“scikit-learn” Available at https://scikit-learn.org/) library of python has also been utilized. The models have been trained on a computer having Ryzen 5, 1600 CPU, 8 GB ram, and Nvidia GeForce 1,050 TI GPU with Linux Mint 19.3 operating system.

Dataset

We have used two publicly available Bangla handwritten character datasets for the experiments—Ekush (Rabby et al., 2019) and BanglaLekha Isolated (Biswas et al., 2017). Primarily, the experiments have been conducted on the Ekush dataset. For comparative analysis purposes, the experiments have also been carried out on the BanglaLekha-Isolated dataset. There are other datasets like ISI (Chaudhuri, 2006), NumtaDB (Alam et al., 2018), and CMATERdb (Sarkar et al., 2012). But they do not have as many classes as Ekush and BanglaLekha-Isolated datasets.

Ekush dataset has the highest number of classes among all Bangla handwritten character datasets. It has 122 classes of characters of four types like modifiers, basic characters, compound characters, and numerals. This dataset contains 734,036 different images. The creators of this dataset have collected handwritten images in a form from 3,086 people among which 50% are male and 50% are female. Table 3 shows the details of the Ekush dataset. On the other hand, the BanglaLekha-Isolated dataset has 84 classes distributed in 50 basic characters, 10 numerals, and 24 compound characters. The images of both datasets are greyscaled images. The images of the Ekush dataset have a fixed dimension of 28 × 28 pixels whereas the dimension of the images of the BanglaLekha-Isolated dataset varies from 36 × 36 pixels to 191 × 191 pixels. During experiments, images from both datasets have been kept with a fixed dimension of 28 × 28.

Table 3 Ekush dataset details.

Character type	No. of classes	No. of instances	
Modifier	10	54,829	
Basic character	50	307,316	
Compound character	52	306,231	
Digit	10	61,374	
Total	122	729,750	

In Fig. 8A, few vowel modifiers from the Ekush dataset are displayed. The vowel modifiers are labeled from 0 to 9 in the dataset. These characters cannot be found isolated. They are found before or after, or both before and after consonants or compound characters. These are the representative or short forms of actual vowels. There are eleven vowels in the Bangla language. They are labeled from 10 to 20. These characters can be found in a word independently. But if they are to be used as phonetics, their corresponding vowel modifiers work on the consonants and compound characters. Few of the vowels are displayed in Fig. 8B. In Fig. 8C, a few of the 39 consonants of the Bangla language are displayed. They are labeled from 21 to 59. A few of the 52 compound characters of the Ekush dataset are shown in Fig. 8D. They are labeled from 60 to 111. Compound characters in Bangla can be constructed by joining two or more consonants. Finally, in Fig. 8E, four of the ten Bangla handwritten digits of this dataset are shown. They are labeled from 112 to 121.

Figure 8 A few representative images of each character type from the Ekush dataset.

(A) A few Bangla handwritten modifiers, (B) a few Bangla handwritten vowels, (C) a few Bangla handwritten consonants, (D) a few Bangla handwritten compounds, and (E) a few Bangla handwritten digits.

Training, testing and validating

The datasets have been divided into three sets—train, validation, and test. The train set has approximately 75%, the validation set has approximately 10%, and the test set has approximately 15% of the data. For the Ekush dataset, 547,131 of 729,750 images in the whole dataset are used for training. And for validation and test, the number of images is 72,842 and 109,777, respectively. For the BanglaLekha-Isolated dataset, for training convnets, 224, 211 images have been used. And in the validation and test sets, there have been 24,881 and 33,310 images, respectively. For training the first level classifiers of the stacked generalization model, the training set has been divided into ten folds using 10-fold cross-validation and a classifier on each fold has been fitted. After combining the first-level classifiers, the second-level classifier has been trained with the validation set of the original dataset.

The reason behind not using the training set again is that the first-level classifiers have already known the training set. That is why the original validation set which is completely unknown for the first level classifiers has been used. After training the second level classifier, it has been tested against the original test images – the same test images that have been used for testing throughout the study. For the bootstrap aggregating method, train and validation sets have been combined. From the combined set, ten bags of datasets have been sampled with replacement using the bootstrap aggregating method. After fitting a convnet on each of the bags, the final classifier (based on the results of ten convnets) has been tested with the original test set.

Optimizers, loss function, batch size and evaluation metrics

To select appropriate optimizers, few models have been trained with various optimizers. The optimizer that has the best performance has been used for model training. Similarly, pre-experiments with various batch sizes have also been conducted and the best performing batch size has been used for actual model creation. As the models have to accomplish a multi-class classification task, the categorical cross-entropy loss function has been used. Moreover, four evaluation metrics have been used throughout the whole working process of this study—precision, recall, F1-score, and accuracy.

Results

The overall performances of the experiments are generally well and competitive to other existing works. In the previous section, the three stages of the experimental setup have been described. In the first stage, the different convnets have demonstrated results ranging from 96.25% accuracy to 97.81% accuracy on the Ekush dataset. For the BanglaLekha-Isolated dataset, accuracy has varied from 88.88% to 93.55%. The most accurate performance that has been recorded for Ekush dataset is 98.68% in terms of accuracy. This has been obtained by the stacked generalization ensemble method. For the BanglaLekha-Isolated dataset, the highest achieving performer has been the bootstrap aggregating method with 93.55% test accuracy. Apart from the classifiers’ performances, other aspects are needed to be discussed. In this section, the results of the study along with various aspects are presented elaborately with appropriate discussion.

Result comparison

The result of this study has outperformed most of the existing works in the domain of Bangla handwritten character classification. In Table 4, a comparison of this work with other existing works has been presented after an extensive literature review. The comparison has proven that our work has exemplary and outstanding results in terms of better performance and classifying Bangla handwritten characters on a large scale. Most of the current works have classified less than 122 characters. Very few of them have classified 122 or more than 122 Bangla handwritten characters. Moreover, this study has beaten (Azad Rabby et al., 2018) which has the same dataset as ours. They have the same train-test split as our experiments with 15% images in the test set. They have reported a 97.73% test accuracy obtained by their EkushNet architecture whereas our stacked generalization method has been able to achieve 98.68% test accuracy. Our developed bootstrap aggregating method has also outperformed the performance of the EkushNet model.

Table 4 Performance comparison with state-of-the-art works on Bangla handwritten character recognition.

Work	Dataset	Number of characters	Method	Accuracy
(%)	
Pal, Wakabayashi & Kimura (2007)	Own collected dataset.	138	MQDF	85.90	
Rahman et al. (2015)	Own prepared dataset with 20000 samples	50	Convnets	85.96	
Purkaystha, Datta & Islam (2017)	BanglaLekha Isolated (Biswas et al., 2017)	80	Convnets	88.93	
Bhattacharya et al. (2016)	Own collected dataset.	152	SVM	94.3	
Alif, Ahmed & Hasan (2018)	CMATERdb (Sarkar et al., 2012)	73	Modified ResNet18	95.10	
Rahaman Mamun, Al Nazi & Salah Uddin Yusuf (2018)	NumtaDB (Alam et al., 2018)	10	Ensemble	96.69	
Azad Rabby et al. (2018)	Ekush (Rabby et al., 2019)	122	Convnets	97.73	
Proposed method	Ekush (Rabby et al., 2019)	122	Stacked Generalization	98.68	
Proposed method	Ekush (Rabby et al., 2019)	122	Bagging	98.37	
Proposed method	BanglaLekha Isolated (Biswas et al., 2017)	84	Stacked Generalization	92.67	
Proposed method	BanglaLekha Isolated (Biswas et al., 2017)	84	Bagging	93.55	

Bhattacharya et al. (2016) and Pal, Wakabayashi & Kimura (2007) have classified 152 and 138 handwritten characters, respectively but have not gained performance like ours. We have achieved a maximum of 98.68% accuracy for the classification task while the other works’ performances have not exceeded 98% accuracy. To further validate our obtained results, the proposed methods have also been applied on the BanglaLekha Isolated dataset (Biswas et al., 2017) which has the second-highest number of Bangla isolated characters among all publicly available datasets in this domain. Our proposed models on this dataset have also outperformed other works. Stacked generalization and bootstrap aggregating methods have yielded 92.67% and 93.55% test accuracy whereas (Purkaystha, Datta & Islam, 2017) have obtained 88.93% test accuracy. They also have not considered all 84 classes of the dataset rather they have built their model on 80 classes which further proves the superiority of our methods as all 84 classes have been used for classification in our work.

Performances of convnets

During training, after each epoch, the training and validation accuracies of the convnets with their respective losses have been observed to check on the stability of models. The models have been trained for a hundred epochs, and at the end of each epoch, they have been validated against the validation images. In Fig. 9, the validation accuracies over the number of epochs for each convnets for the Ekush dataset have been compared. The figure depicts that all seven models have generally smooth learning curves. The comparison of the validation accuracies of the models indicates that the ResNet50 model has been better than other models. And, the DenseNet model has been worse than others. The Xception model has a performance almost like the ResNet50, and AlexNet has a performance close to DenseNet. The other three models – small CNN, VGG16, and VGG19 are somewhere in between good and bad.

Figure 9 Validation accuracy vs epochs comparison for the convnets on the Ekush dataset.

On the other hand, Fig. 10 depicts the validation loss over the epochs of each convnet for the Ekush dataset. Unlike validation accuracy curves, the validation loss curves of convnets are not smooth. The validation loss of the small CNN model is lesser than other models in a certain epoch while DenseNet has a higher loss than others. The small CNN and DenseNet models have minimum losses of 0.0459 and 0.1841, respectively. These experimental values indicate that the small CNN model has been more stable than other models in terms of validation loss, and the DenseNet model has been more unstable than others. The other models have stability in between the two discussed models. Moreover, the ResNet50 model has outperformed all the other six models with 97.81% test accuracy. It has also 97.82%, 97.81%, and 97.81% precision, recall, and F1-score, respectively.

Figure 10 Validation loss vs epochs comparison for the convnets on the Ekush dataset.

All the convnets’ precision, recall, F1-score, and accuracy on the test set of the Ekush dataset are given in Table 5. From there, it can be observed that the Xception model has come close to ResNet50 with 97.63% accuracy, 97.64% precision, and 97.63% recall. While these two models have performed better, VGG16, VGG19, and small CNN models have average performances with accuracies of 96.97%, 97.05%, and 97.30%, respectively. On the contrary, DenseNet and AlexNet have comparatively poor performances than others with accuracies of 96.25%, and 96.80%, respectively. The reason behind the good performance of ResNet50 and Xception is that they have very deep and complex structures. These architectures have been able to extract the patterns of Bangla handwritten characters efficiently. It is also to be noted that the poor-performing models such as DenseNet and AlexNet have also taken more time to train than the models with better performance. They have also been slower than the better-performing models like ResNet50 and Xception in the testing phase. The performances of convnets on the BanglaLekha-Isolated dataset are also presented in Table 5. Among the convnets, the ResNet50 model has been the top performer with 92.63% test accuracy while the AlexNet model has yielded only 88.88% accuracy.

Table 5 All models’ performances on the Ekush and BanglaLekha-Isolated datasets.

Dataset	Methods	Models	Precision (%)	Recall (%)	F1-score (%)	Accuracy (%)	
Ekush Dataset	Convnets	AlexNet	96.80	96.80	96.79	96.80	
VGG16	97.06	97.05	97.04	97.05	
VGG19	96.99	96.97	96.97	96.97	
ResNet50	97.82	97.81	97.81	97.81	
Xception	97.64	97.63	97.63	97.63	
DenseNet	96.31	96.25	96.26	96.25	
Small CNN	97.33	97.30	97.30	97.30	
ResNet50 as the feature extractor	Logistic Regression	97.76	97.76	97.76	97.76	
SVM	97.76	97.75	97.75	97.75	
Naïve Bayes	97.2	97.15	97.16	97.15	
Decision Tree	95.75	95.74	95.74	95.74	
Ensemble	Stacked Generalization	98.69	98.68	98.68	98.68	
Bootstrap Aggregating	98.38	98.37	98.37	98.37	
Adaboost	96.42	96.36	96.36	96.37	
Xgboost	96.59	96.58	96.58	96.58	
Random Forest	97.33	97.32	97.31	97.32	
BanglaLekha-Isolated Dataset	Convnets	AlexNet	88.99	88.88	88.89	88.88	
VGG16	92.16	92.11	92.10	92.11	
VGG19	89.86	89.75	89.76	89.75	
ResNet50	92.68	92.63	92.65	92.63	
Xception	90.37	90.19	90.22	90.19	
DenseNet	89.60	89.41	89.50	89.41	
Small CNN	92.63	92.59	92.58	92.59	
ResNet50 as the feature extractor	Logistic Regression	92.24	92.17	92.19	92.17	
SVM	92.09	92.03	92.04	92.03	
Naïve Bayes	91.87	91.67	91.72	91.67	
Decision Tree	90.12	90.04	90.06	90.04	
Ensemble	Stacked Generalization	92.78	92.67	92.67	92.67	
Bootstrap Aggregating	93.60	93.55	93.55	93.55	
Adaboost	91.72	91.62	91.66	91.62	
Xgboost	91.92	91.86	91.88	91.86	
Random Forest	92.28	92.20	92.22	92.20	

Performances of ResNet as feature extractor

In the second phase of the study, the best performing convnet has been used as a feature extractor for the classification task. The goal of this experiment has been to explore the applicability of regular classification algorithms in image classification. As it is known that the images are subject to contextual information. Therefore, simply passing the original pixel values of images into a classifier without giving additional information would not yield better performance. To use those classifiers, contextual features from images are needed to be extracted. For feature extraction purpose, the best-performing convnet has been used by removing its classification layers—in this case, that is ResNet50. As a convnet has been used, the performance of the classifiers depends on how well the convnet can extract features from images. Better convnet will help the classifiers to obtain better performance.

In Table 5, for both datasets, the performances of logistic regression, support vector machine, naïve Bayes, and decision tree classifiers with pre-trained ResNet50 as feature extractor have been presented. Among them, the SVM classifier performed better than other classifiers with 97.75% accuracy, 97.76% precision, and 97.75% recall on the Ekush dataset. On the other hand, logistic regression has been the top performer on the extracted features with 92.17% accuracy, 92.24% precision, and 92.17% recall for the BanglaLekha-Isolated dataset. Although the shallow classifiers could not beat the original ResNet50 classifier which has 97.81% accuracy, they have worked exceptionally well with a very small feature set for the Ekush dataset. The same minor downgrade in performances of shallow classifiers from the original ResNet50 model has also been observed for the BanglaLekha-Isolated dataset.

Results of ensemble methods

In the final stage of the experiments, few ensemble methods have been employed for the classification task. The ensemble methods are broadly categorized into two types. One type of ensemble method is where the classifiers have been built from scratch on the original training images, and the other type is where the classifiers have been trained on the training set with extracted features by the ResNet50 model. The classifiers of the first type of ensemble method have comparatively better performances than the second. Even they have demonstrated better performances than all the other methods employed during this study.

Performance of stacked generalization ensemble method

To create the second-level ensemble classifier, ten different classifiers have been trained initially. After training the first-level classifiers, they have been tested with the original test images. In Fig. 11, their performances on the Ekush dataset are presented. The performances of first-level classifiers have a resemblance with their original convnets. ResNet50 and Xception models have better performances than others, and AlexNet has the lowest performance. The best first-level classifier has been the ResNet50 model that has been trained on the 7th fold with rotation range = 14, height shift range = 0.10, width shift range = 0.10, and zoom range = 0.10. It has a test accuracy of 97.90% which is better than its ResNet50 counterpart trained on the whole training set. This better-performing trend only has been observed in the ResNet models.

Figure 11 First and second level classifiers’ performances on Ekush dataset.

The other models have lost some performance for being trained on a subset of the original training set. The stacked generalization model has outperformed the other individual classifiers with 98.68% test accuracy. Using different models, different image augmentations, and a different subset of the training set on each first level classifier has helped to achieve better performance of the second level classifier. Precision, recall, F1-score, and the accuracy comparison of first and second-level classifiers on the Ekush dataset are given in Fig. 11. From the figure, it can be seen that the stacked model has the highest performance in all the evaluation metrics. The stacked model has 98.69% precision, 98.68% recall, 98.68% F1-score, and 98.68% accuracy on the Ekush dataset while the first level models do not have the same level of performance. On the other hand, from Table 5, the stacked model has achieved 92.78% precision, 92.67% recall, 92.72% F1-score, and 92.67% accuracy on the BanglaLekha-Isolated dataset.

Performance of bootstrap aggregating ensemble method

Like the stacked generalization method, better performing convnets have been utilized to create ten individual classifiers. Each of them has been trained on one of the ten bags of training images. Figure 12 shows the test performances of these ten models on the Ekush dataset. Like the first-level classifiers of the stacked generalization method, the individual classifiers of the bootstrap aggregating method have similar performances. ResNet50 model with image augmentation of rotation range = 10, height shift range = 0.1, width shift range = 0.1, and zoom range = 0.1 has been the better performer than others.

Figure 12 Individual and bagged classifiers’ performances on Ekush dataset.

AlexNet model has worse performance than other models. It can also be seen from the figure that ResNet18, VGG16, and Xception models have come close to ResNet50 in terms of test accuracy. After finishing the training of these convnets, the test images have been tested using the majority voting method on the output of ten convnets. The performance of the predictions based on ten models has outperformed all the individual models. For Ekush dataset, the evaluation metrics-wise comparison of individual models and bagged models has been shown in Fig. 12 as well. The figure depicts that the bagging method has yielded 98.38%, 98.37%, 98.37%, and 98.37% precision, recall, F1-score, and accuracy, respectively. On the other hand, the individual models do not have any value of evaluation metrics larger than 97.5%. This increase in model performances of both stacked generalization and bagging methods justifies the efficiency of these classification methods.

The bootstrap aggregating method has also obtained superior performances than individual convnets for the BanglaLekha-Isolated dataset. From Table 5, it can be said that the bagging method has achieved 93.60% precision, 93.55% recall, 93.55% F1-score, and 93.55% accuracy. In fact, the bagging method has achieved the best result among all the other experiments on the BanglaLekha-Isolated dataset.

Performance of boosting and random forest

Among the other ensemble classifiers, the random forest has performed better than the other two for both datasets. The precision, recall, F1-score, and accuracy of these models are demonstrated in Table 5. The random forest has secured first place with 97.32% accuracy, 97.33% precision, and 97.32% recall on Ekush. On the other hand, AdaBoost has worse performance than XGBoost with 96.37% accuracy, 96.42% precision, and 96.36% recall while XGBoost has 96.58%, 96.59%, and 96.58% accuracy, precision, and recall, respectively on the Ekush dataset.

Ablation studies

Batch size and optimizers are two important parameters when it comes to large-scale image classification. As the task of image classification is very computationally expensive, choosing the appropriate batch size and optimizer can play significant roles in model performance. For this purpose, a few pre-experiments with batch size and optimizers have been carried out. For choosing a batch size, a small CNN model has been developed and trained for 40 epochs with various batch sizes. From the experiments, it has been observed that a larger batch size usually helps the model to obtain better performance in terms of test accuracy. Experiment with 1,024 batch size has yielded better performance than that of 512 batch size on Ekush dataset. The larger batch sizes do not always yield better performance either. Classifiers with 1,536 or 2,048 batch sizes have performed less accurately than that of 1,024 batch sizes.

In the second ablation study, a ResNet50 model has been developed and trained for ten epochs with five optimizers—adam, RMSprop, stochastic gradient descent (SGD), Adagrad, and Adadelta. Among different optimizers, SGD has performed worse than other optimizers in terms of validation accuracy (89.52%). RMSprop has higher validation accuracy than others (96.48%). As the batch size of 1024 and RMSprop optimizer have better performance in the pre-experiments, this combination of batch size and optimizer has been used for the rest of the experiments.

Time is another important factor that must have to be considered when it comes to model evaluation. Although training time is often overlooked if the model can classify an instance faster in a real-world environment. But due to a lack of available computational resources, training a convnet can become time-consuming. With the experimental setup of this study, the VGG19 model has taken on an average 0.3271 millisecond time to process an image while training. This is the lowest time among the seven convnets. In Table 6, the time to process an image by each model, and time to predict an image are given. It can be seen in the table that the DenseNet model has taken more time than the other models to process an image during training. AlexNet has also taken a comparatively longer time to be trained.

Table 6 Training and testing time of different models on the Ekush dataset.

Model	Time to process an image during training (millisecond)	Time to predict an image during testing (millisecond)	
AlexNet	0.9766	0.3672	
Small CNN	0.3916	0.1582	
DenseNet	1.468	0.4307	
ResNet50	0.4727	0.2031	
VGG16	0.332	0.1572	
VGG19	0.3271	0.336	
Xception	0.5469	0.1914	

On the other hand, the VGG16 model has the lowest average prediction time per image. It takes 0.1572 milliseconds to classify an input image. Besides, the DenseNet model needs more time than any other model to predict an image (0.4307 ms). Another interesting thing is that all the models tend to work faster in the testing phase than in the training phase except for VGG19. VGG19 has taken 0.336 milliseconds to test an image while it has taken 0.3271 milliseconds to process an image during training. The most significant improvements have been seen in the time of DenseNet to predict. It predicts an image in 0.4307 milliseconds which is 1 millisecond less than the training time per image. The other models have shown improvements in testing time too. However, in terms of performance-time trade-off, ResNet50 and Xception have been better classifiers. They have achieved better test accuracy than other models with relatively faster predictions. A point to be noted that all the ablation experiments have only been applied on Ekush dataset under the assumption that a similar pattern of performances will be found for the BanglaLekha-Isolated dataset.

Discussion

Among the individual convnets, the ResNet50 model has the best performance for the Bangla handwritten recognition task for both datasets. And among the regular classifiers using ResNet50 as a feature extractor, the SVM classifier has the best performance for the Ekush dataset while the logistic regression classifier has the best performance on the BanglaLekha-Isolated dataset. And, finally, among the ensemble methods, stacked generalization has the best performance for the Ekush dataset while the bootstrap aggregating model has the best performance on the BanglaLekha-Isolated dataset. Among all the classifiers, the stacked generalization and bootstrap aggregating methods have produced the best classifiers for two datasets, respectively. Two of the seven convnets have performed with the best accuracies on the primary dataset of our experimental setup. Those convnets are the ResNet50 and the Xception network. Similarly, two of the ensemble methods have the highest performances. Stacked generalization ensemble and bootstrap aggregating methods have been proven to be exemplary in the field of Bangla handwritten recognition. These ensemble methods have not been used in this domain-specific classification task before. Additionally, the convnets have been used as feature extractors. Moreover, various experiments have been carried out to tune the different aspects related to this study. These types of classifiers can be used for handwritten automation in various real-world applications.

In the first phase of the study, on the primary dataset, the reason for ResNet50 and Xception models having better results is their complex and sophisticated structures. For having more than 120 classes in the dataset, and for the complexity of various handwritten Bangla characters, the classification requires more robust and advanced architectures. As mentioned, these two models have more robust and deeper architectures than most of the other convnets that have been used, they have been able to yield outstanding performances. The training time has also been faster for them as they utilize their shortcut connections. Another aspect is to discuss the excellent performances of stacked generalization ensemble and bootstrap aggregating methods. To build classifiers using these two methods, individual convnets have been trained separately. When the final classifier predicts an image into a specific class, it uses all those bits of knowledge learned from several convnets instead of a single model. Those several convnets are also trained differently one from another with different architectures and with different image augmentation. Using these wide ranges of model learnings has allowed stacking and bagging methods to perform exceptionally better than single convnets.

However, the other ensemble methods have not performed as good as stacking and bagging. For the other ensemble methods, the dataset after extracting features using ResNet50 has been used. The performance generally relies on the feature extraction process. Better feature extraction helps to yield better model performances. As the other ensemble classifiers have been trained on the features extracted by ResNet50, it is obvious that they will not be able to beat the original ResNet50 model. A similar justification is applicable for worse performances by logistic regression, SVM, naïve Bayes, and decision tree classifiers. Nevertheless, achieving a respected result with fewer features (80 instead of 1,024) is a positive outcome.

Misclassification

After testing the convnets, their confusion matrices and class-wise classification performances have been observed. In this large-scale classification, the classifiers have performed poorly on a few classes from 122 classes of Bangla handwritten characters of the Ekush dataset. All convnets have demonstrated a similar trend in misclassifying instances from a few specific classes. In Table 7, a few examples of classes are presented with their supports, number of false positives, and number of false negatives, where the ResNet50 classifier has shown poor, average, and good performance, respectively.

Table 7 False positive, false negative of a few classes of ResNet50 model for Ekush dataset.

Performance	Class	Support	Number of false-negative	Number of false positive	
Poor	111	156	47	32	
46	894	60	60	
84	872	63	53	
97	642	48	53	
Average	103	924	46	2	
16	926	15	15	
39	926	42	46	
Good	20	928	5	2	
96	926	3	6	

There are some reasons behind failing to perform better for some specific classes. One of them is the lack of a sufficient number of training and testing images in a class. For example, the handwritten character labeled as 111 has fewer training and testing instances. The lower support value for this class from Table 7 indicates the insufficiency of the number of instances during testing. This class has only 1,237 training images, while the other classes have more than 4,500 images each on average. Although, this is not only the reason for poor performance. Similar patterns of Bangla handwritten characters across different classes are also responsible. There are few characters in Bangla that have almost the same pattern. A few of the examples are given in Fig. 13. Character labeled as 111 has similarity with character labeled as 69, class 19 has similarity with class 84, and so on. These close resemblances between two classes make a classifier predict one class into another. The empirical data also supports that. Among the 47 false-negative instances for class 111, 23 of them are misclassified as class 69. And among 32 false-positive instances, 15 of them are from class 69. Similar misclassifications have happened for other classes that are presented in Table 7 under the “poor” category. For having similar patterns, people often mistake one character for another. Thus, this results in writing them wrongly. For this reason, there are also some wrongly labeled images in the dataset. For example, from Fig. 13, the image from the left labeled as 19, and the corresponding image from the right labeled as 84 are the same characters but labeled as different classes. Moreover, for those that are categorized under average and good performance, the ResNet50 classifier does not have a majority portion of misclassified instances that are in a specific class, rather the misclassified instances are evenly distributed into different classes.

Figure 13 A few misclassified instances from Ekush dataset by ResNet50 model.

Conclusion and Future Works

Through various experiments, the applicability of convolutional neural network-based ensemble learning methods for handwritten character recognition has been proved. The empirical evidence shows that predictions are more accurate with the models that combine results of multiple convnets than a single convnet. Single convnets may not have the best performance of this work but the robust architectures among them surely have demonstrated impressive results. The ResNet50 and the Xception models have standout performances in the image recognition tasks. The outstanding results that have been attained are also among the top performances that have been reported in the domain of Bangla handwritten character recognition. Another significant aspect is to classify more than 120 handwritten characters. Recognizing them on such a large scale with such good performance is the major contribution of this work.

However, there are some limitations associated with this study. Only six of the popular convolutional neural networks have been used. The other convnets like Inception (Szegedy et al., 2015), FractalNet (Larsson, Maire & Shakhnarovich, 2019), NiN (Lin, Chen & Yan, 2014), etc. can also be used for classification tasks. Additionally, the training of models from scratch can be a tedious job. Exploring the applicability of the pre-trained transfer learning models in the classification task can be a new research direction. Bangla is a very rich language. There are more than 200 compound characters in it. Among them, only 52 compound characters are classified in this study. There is a lack of an available compound character dataset. A dataset with more compound characters can be curated. In Ekush (Rabby et al., 2019) dataset, there are few wrongly labeled images. Those are needed to be labeled properly. Additionally, only five of the ensemble methods have been used for the classification challenge. The other ensemble methods can be applied for this classification task as well. An important aspect of the ensemble methods is that for boosting and random forest, the reduced dataset has been used. For this reason, the expected performance has not been achieved. To achieve better performance, the classifiers using these methods are needed to be trained from scratch directly on the training images like stacked generalization and bagging ensemble methods. Moreover, the applicability of developed classifiers in this study is only limited to isolated handwritten characters. The systems that can recognize words and sentences from images are needed to be created. Extracting words and sentences from images is also very important. Those systems can help us to obtain true autonomous experience in the field of handwriting recognition and extraction.

Additional Information and Declarations

Competing Interests

Author Contributions

Data Availability

The authors declare that they have no competing interests.

Mir Moynuddin Ahmed Shibly conceived and designed the experiments, performed the experiments, analyzed the data, performed the computation work, prepared figures and/or tables, authored or reviewed drafts of the paper, and approved the final draft.

Tahmina Akter Tisha conceived and designed the experiments, performed the experiments, analyzed the data, performed the computation work, prepared figures and/or tables, authored or reviewed drafts of the paper, and approved the final draft.

Tanzina Akter Tani conceived and designed the experiments, analyzed the data, authored or reviewed drafts of the paper, and approved the final draft.

Shamim Ripon conceived and designed the experiments, prepared figures and/or tables, authored or reviewed drafts of the paper, and approved the final draft.

The following information was supplied regarding data availability:

The data and code are available at figshare:

Shibly, Moynuddin Ahmed (2021): Codes. figshare. Journal contribution. DOI 10.6084/m9.figshare.13578173.v2.

Shibly, Moynuddin Ahmed (2021): Dataset. figshare. Dataset. https://figshare.com/articles/dataset/Dataset/13577873.

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
