# Peer review of "Convolutional neural network-based ensemble methods to recognize Bangla handwritten character"

_PeerJ Computer Science, doi:10.7717/peerj-cs.565_

## Round 0.1 · original submission · Major Revisions

The reviewers have raised a number of critical comments on this paper. The major concern is to improve the presentation of the proposed method and the experiments. In addition, please make sure the contrbution of the method is clearly discussed in the revised version.

·

Basic reporting

There are too many long paragraphs, and the text is not aligned at both ends, which affects reading.

Experimental design

The experimental design is good, and a large number of experiments have been conducted to verify the model.

Validity of the findings

The curves of images 3 and 4 are not very distinguishable, and the viewpoints mentioned in the article are not well drawn.
Table 8 for comparison with other studies hopes to be sorted for easy comparison.

Additional comments

Very meaningful work, but some details need to be revised. Great work!

Reviewer 2 ·

Basic reporting

This paper explores new methods of using ensemble on top of CNN-based feature extraction to improve Bangla character recognition (or classification in machine learning terms). The authors employ datasets used by previous work in literature and obtain better results as reported, with more classes considered. This work, along with others that consider other languages are very good contributions to many communities. In this first round of review, however, several flaws in the methods and experiments, if not also in others, are identified as below, and improvements are suggested to be made to make the paper better and more convincing.


English and format: While the use of English throughout the manuscript is professional, I suggest changing the following:
- Use of enumeration without using “etc”, maybe by replacing it with “such as”.
- Line 84, change to the format as instructed: (Smith, 2001a; Smith, 2001b)
- Line 481, use 500,000 instead of 500000. Similarly for lines 465 and 466, and other places with big numbers.
- Inconsistent uses of metric names, such as F1 in line 506 and f1 in 515 or in other places.


Introduction: while the authors provide much information, some suggestions to improve are made as follows:
- Reference for the first paragraph should be made about how significant Bangla is
- When stating Bangla is more difficult than other languages (which is a very strong claim indeed), the authors should present the comparison clearly, not assuming by looking at only the statistics of Bangla, readers should be able to agree. Such claims might or might not be true with various readers, for example, those are from English-, Chinese-, or Arabic-speaking countries.
- Another suggestion for comparison is to present a figure to illustrate how complex Bangla characters can be.


Literature:
- References should be added for well-known CNN architectures such as Resnet50 (from Kaiming He).
- Literature review does not clearly state the position of this paper: how this paper, in particular, is different from or similar to the previous work. This is important because it reinforces the contribution.
- When it comes to classifying characters only, this work is similar to the vast literature of recognizing, say English characters using HMM or LSTM with IAM, SD19, or other datasets. See for example https://arxiv.org/pdf/2008.08148.pdf. So I suggest the authors make a little review along this line of work to connect more of your paper to the community.

Experimental design

Methods (Section 3):
- Your model is very important. As a result, I suggest you describe the motivation as to why you choose to use such components, e.g, why CNN, why specific ensemble methods? Likewise, the authors need to answer many questions “why” about the choices of architectures.
- After that, the authors can describe the architecture in detail.
- However, sections 3.2 and 3.3 should be purged because they are common knowledge
- Section 3.1 should be placed in section 4 (at the beginning of it). Section 3 should be devoted to describing the model.
- Sections 3.5.1 and 3.5.2 have very long enumerations of components. Maybe the authors should trim it down to a small paragraph each or a small table each.


The research question of Bangla character recognition is useful and well-defined. However, there are the following suggestions to improve:
- First, section 3.1 should be moved to here to describe the datasets. Please state clearly which datasets you are using. For each such dataset, you need to compare it with previous work using it (at least one state-of-the-art baseline).
- Section 4.1 is not clear because the authors refer to only 1 dataset.
- References to Tensorflow/Keras need to be made.
- The use of PCA seems not well-motivated. This linear method would probably make the extracted features lose some information. To answer this question clearly, the authors should investigate the use of current architecture with and without PCA (e.g. in an ablation study). The reasons are two-fold:
1/ The drawback of PCA as said. Why don’t you use an FC layer to narrow down the classes, e.g. 2048 -> 1024 -> 112 as in normal usages of CNN? This is probably the easiest and most straightforward.
2/ Can PCA be implemented end-to-end along with your model, and how much slower it could be when combining with CNN? Usually, with high dimensional data, PCA is slow.
- Section 4.3, the formula in lines 513 to 516, and descriptions of related metrics are common knowledge and should be trimmed. If needed, make related references to those metrics.
- Table 8 is probably the most important result to present and should be made first and foremost. Other ablation studies such as with different architectures, optimizers, methods should be placed later.
- However, the results in Table 8 are questionable because it is needed to make a fair comparison between your methods and others, based on the same set of datasets (including splits between train/val/test). Important changes should be made:
1/ Compare your methods with others on their datasets (at least one good baselines, explain why you choose the baseline(s))
2/ Compare other methods with your current datasets (again, with a good baseline).

Validity of the findings

The authors employ widely used components of CNN, Ensemble, and PCA. However, they claim their combination is novel, which was made clear in the literature review. Nonetheless, they need to motivate their choices more convincingly. Experimental results should be made more clearly and in a more convincing way as suggested below.

Dataset: The authors adhere to PeerJ regulations by providing citations to datasets being used. However, the descriptions of datasets lack important characteristics such as:
- the resolutions and the format (RGB or not)
- the authors enumerate Bangla datasets without stating directly which datasets they are going to use in specific, which they should heavily focus on

Additional comments

The research question of Bangla character recognition is useful for the community and well-defined. However, your choices of architectures, their descriptions, and experiments should be further improved to make the paper more convincing.

---

## Round 0.2 · accepted · Accept

The comments are well addressed.

·

Basic reporting

The content of the article is clear, but article typesetting is not easy to read and may require post orchestration.

Experimental design

The experiments were well designed and largely validated.

Validity of the findings

The experimental results are clear and unambiguous.

Additional comments

Very meaningful work, but some details need to be revised. Great work!